# Signed Graph Neural Networks: A Frequency Perspective

**Rahul Singh**                                                                                                                                 *rasingh@gatech.edu*
*Machine Learning Center*
*Georgia Institute of Technology, Atlanta*

**Yongxin Chen**                                                                                                                           *yongchen@gatech.edu*
*School of Aerospace Engineering*
*Georgia Institute of Technology, Atlanta*

**Reviewed on OpenReview:** *https://openreview.net/forum?id=RZveYHgZbu*

## Abstract

Graph convolutional networks (GCNs) and its variants are designed for unsigned graphs containing only positive links. Many existing GCNs have been derived from the spectral domain analysis of signals lying over (unsigned) graphs and in each convolution layer they perform low-pass filtering of the input features followed by a learnable linear transformation. Their extension to signed graphs with positive as well as negative links imposes multiple issues including computational irregularities and ambiguous frequency interpretation, making the design of computationally efficient low pass filters challenging. In this paper, we address these issues via spectral analysis of signed graphs and propose two different signed graph neural networks, one keeps only low-frequency information and one also retains high-frequency information. We further introduce magnetic signed Laplacian and use its eigendecomposition for spectral analysis of directed signed graphs. We test our methods for node classification and link sign prediction tasks on signed graphs and achieve state-of-the-art performances.

## 1 Introduction

Graph neural networks (GNNs) learn powerful node representations by capturing local graph structure and feature information (Ma & Tang, 2021; Wu et al., 2021). The existing GNN architectures have focused almost exclusively on graphs with nonnegative edges, which encode some kind of similarity relation between the incident nodes. In contrast, negative edges are often useful to model dissimilarity relations (Kumar et al., 2016; Dittrich & Matz, 2020): for instance, in social networks, users may have common/opposite political views, trust/distrust one another's recommendations, or like/dislike each other. Such dissimilarity relations can be modeled using signed graphs by allowing the edges to take both positive or negative values. In this paper, we are interested in GNN designs for signed graphs.

There exist several methods such as SGCN (Derr et al., 2018), SiGAT (Huang et al., 2019), and SNEA (Li et al., 2020) that consider signed links to learn node embeddings in a task-specific end-to-end manner. These GNN based models are based on structural balance theory for signed graphs (Heider, 1946; Cartwright & Harary, 1956). In this work, we propose an alternative solution to GNNs for signed graphs from a frequency perspective via spectral domain analysis. Recall that the spectral domain analysis of unsigned graphs has been widely used to develop GNN architectures. Many well-known GNNs including spectral-GNN (Bruna et al., 2014), ChebNet (Defferrard et al., 2016), GCN (Kipf & Welling, 2017), AGCN (Li et al., 2018), and FAGCN (Bo et al., 2021) rely on spectral domain analysis. These designs are based on the frequency interpretation derived from the eigendecomposition of the normalized unsigned graph Laplacian. However, direct application of the existing spectral domain GNN designs to signed graphs is problematic, mainly due to (i) possible zero diagonal entries in the degree matrix making the normalization of the Laplacian prohibitive and (ii) possible negative eigenvalues of the graph Laplacian, making the frequency ordering

Table 1: Different Laplacians and their applicability.

|  | Unsigned | Directed | Signed | Directed Signed |
|---|---|---|---|---|
| Laplacian $\mathbf{L}$ | ✓ | ✗ | ✗ | ✗ |
| Magnetic Laplacian $\mathbf{L}^q$ | ✓ | ✓ | ✗ | ✗ |
| Signed Laplacian $\bar{\mathbf{L}}$ | ✓ | ✗ | ✓ | ✗ |
| Signed Magnetic Laplacian $\bar{\mathbf{L}}^q$ | ✓ | ✓ | ✓ | ✓ |

somewhat ambiguous, i.e., whether the smallest negative, positive, or absolute value, eigenvalues should be used as low frequency (Knyazev, 2017).

To address these issues, we turn to signed graph signal processing (Dittrich & Matz, 2020) which provides frequency interpretation for features lying on the signed graphs, and propose spectral domain signed GNNs based on it. Specifically, we propose two different GNN designs for signed graphs: Spectral-SGCN-I and Spectral-SGCN-II. The former considers fixed low-pass filter keeping only low-frequency information during aggregation process, whereas the later is based on attention mechanism retaining low as well as high-frequency information. Extending these methods to directed signed graphs is another challenge. For handling directed signed graph, we further introduce spectral methods for directed signed graphs. We evaluate the performance of our methods on node classification and link sign prediction tasks on signed graphs. Our contributions are summarized below.

- We present a principled approach to designing graph neural networks for signed graphs based on the spectral domain analysis over signed graphs.

- We instantiate our approach with two graph neural network architectures for signed graphs, one behaves like a low-pass filter and one also retains high-frequency information.

- We introduce signed magnetic Laplacian (see Table 1) for spectral analysis of directed signed graphs and utilize it in feature aggregation process.

- We evaluate our method through extensive evaluations on node classification as well as link sign prediction tasks for signed graphs and achieve state-of-the-art performances.

**Related Work:** There exist many different methods to learn node representations (embeddings) on signed graphs. Most of these methods are derived from the balance theory (Cartwright & Harary, 1956) that was motivated by social networks. The balance theory concept was inspired by the sociological observations that friends (enemies) of my friends are also my friends (enemies), and enemies of my enemies are my friends. Alternatively, balance theory is shown equivalent to a simple assumption that the nodes in a signed graph can be divided into two conflicting groups. The existing signed graph embedding methods can be classified into two categories: signed network embeddings and GNN based methods. Signed network embedding methods including SiNE (Wang et al., 2017), SIDE (Kim et al., 2018), SIGNet (Islam et al., 2018), SLF (Xu et al., 2019), and ASiNE (Lee et al., 2020) learn node representations in a task-independent manner so that nodes connected via positive links are in close proximity to each other, whereas nodes connected via negative links are distant from each other. SiNE (Wang et al., 2017) learns signed network embedding with an objective function guided by the balance theory, while SIDE (Kim et al., 2018) and SIGNet (Islam et al., 2018) use specifically designed random walks to model the balance theory. Moreover, SLF (Xu et al., 2019) also performs two types (i.e., positive and negative) of node embeddings for each target node and takes a combination of the two node embeddings as the final node embedding. These node embeddings are then used for task in hand separately.

GNN based methods including signed graph convolutional networks (SGCN) (Derr et al., 2018), signed graph attention networks (SiGAT) (Huang et al., 2019), signed network embedding based on attention (SNEA) (Li et al., 2020), and are jointly trained to learn node embeddings along with the task in hand in an end-to-end manner. SGCN is a state-of-the-art signed GNN model considering balanced and unbalanced paths motivated from the balance theory to aggregate local graph information with fixed coefficients. SNEA further extended SGCN to incorporate learnable attention coefficients for aggregating balanced and unbalanced paths. SiGAT is a motif-based GNN model to learn the node representation inspired by GAT (Veličković et al., 2018).

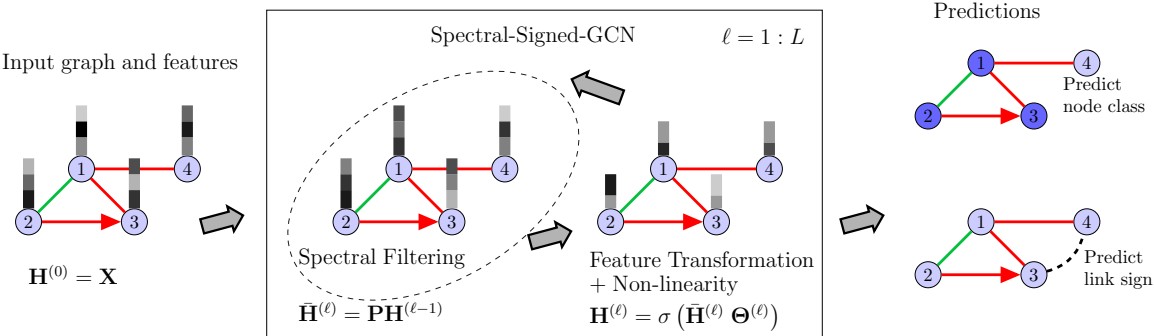

Figure 1: Spectral Signed GNNs. In the signed graph, Red color edges represent negative links and Green color edges represent positive links. The spectral signed GCN filters and transforms the features repeatedly throughout $L$ layers and then applies a linear prediction.

These methods are not applicable to directed signed graphs. SDGNN (Huang et al., 2021) is a recent work applicable to signed directed graphs based on balance and status theory. SSSNET (He et al., 2022b) is another GNN based work with a focus on the clustering of signed graphs. Recently, after this work was completed, we came to know about a concurrent work Magnetic Signed GNN (MSGNN) (He et al., 2022a) that also utilizes a spectral domain approach. The focus of it is on directed signed graphs.

For balanced graphs, the eigenvectors of the signed Laplacian follow certain properties as analyzed in (Dittrich & Matz, 2020). However, it is difficult to relate the signed spectral analysis with balance theory in case of unbalanced graphs. It is an interesting problem to explore the relationship between balance theory and spectral analysis for unbalanced graphs, which is out of scope of this work.

## 2 Problem Definition

Given a (directed) signed graph along with input node features, the goal is to learn low-dimensional latent node features (embeddings). These features are subsequently used for a specific task such as node classification and link sign prediction. Let $\mathcal{G} = (\mathcal{V}, \mathcal{E}^+ \cup \mathcal{E}^-)$ be a directed signed graph, where $\mathcal{V}$ is the set of $N$ number of nodes, $\mathcal{E}^+$ is the set of directed positive edges, and $\mathcal{E}^-$ is the set of directed negative edges. The adjacency matrix of the graph is denoted as $\mathbf{A} \in \mathbb{R}^{N \times N}$ and has entries from $\{+1, -1, 0\}$. If $(i, j)$ is not an edge of the graph, then the corresponding entry is $\mathbf{A}(i, j) = 0$. $\mathbf{A}(i, j) = +1$ denotes a positive edge from node $i$ to $j$, whereas $\mathbf{A}(i, j) = -1$ denotes a negative edge from node $i$ to $j$. The input node features are represented as a matrix $\mathbf{X} \in \mathbb{R}^{N \times F}$ with $\mathbf{x}_i \in \mathbb{R}^N$ (column of $\mathbf{X}$) representing the $i^{th}$ feature channel of $\mathbf{X}$ and $F$ denotes the total number of feature channels.

As depicted in Figure 1, in each GCN layer the input features are first filtered and then (linearly) transformed followed by a non-linear activation. In the $\ell^{th}$ GCN layer, the linear transformation part contains learnable parameters $\mathbf{\Theta}^{(\ell)}$. After $L$ number of GCN layers, the latent features are fed to the prediction head, which can be designed to either predict node labels or predict link signs. For both node label prediction (node classification) and link sign prediction tasks, cross-entropy loss can be used based on the available labels and signs, respectively. The spectral filtering operation in each signed GCN layer is the main focus of this work.

## 3 Preliminaries

Popular spectral domain designs of graph neural networks including ChebNet (Defferrard et al., 2016), GCN (Kipf & Welling, 2017) and their further improvements such as AGCN (Li et al., 2018), Simplified GCN (Wu et al., 2019), and FAGCN (Bo et al., 2021) are based on the spectral analysis of signals (features) defined on an unsigned graph. The spectral analysis of graph signals has been studied under the umbrella of the graph signal processing (GSP) framework (Shuman et al., 2013; Ortega et al., 2018; Manoj et al.,

2018). When the graph is signed, these existing spectral GNN methods encounter multiple challenges. In this section, we briefly discuss how spectral domain analysis of graphs is used for spectral domain GNN designs and point out the issues when dealing with signed graphs.

### 3.1 Graph Fourier Transform and Spectral Filtering

GSP is concerned with the generalization of classical signal processing concepts and tools to graph signals. GSP relates the vertex and spectral domains of a graph, much as classical signal processing connects the time and frequency domains of a time series (Cheung et al., 2020). The eigenvalues and eigenvectors of the (unsigned) graph Laplacian provide a notion of frequency for signals defined on a graph. The graph Laplacian eigenvectors associated with low frequencies, vary slowly across the graph, i.e., if two vertices are connected by an (unsigned) edge, the values of the eigenvector at those locations are likely to be similar. The eigenvectors associated with larger eigenvalues oscillate more rapidly and are more likely to have dissimilar values on vertices connected by an edge. The graph Fourier transform and its inverse give us a way to equivalently represent a signal in two different domains: the vertex domain and the graph spectral domain.

The graph Fourier (spectral) analysis relies on the spectral decomposition of graph Laplacians. The traditional combinatorial graph Laplacian is defined as $\mathbf{L} = \mathbf{D} - \mathbf{A}$, with $\mathbf{D} = \mathrm{diag}\{d_1, d_2, \ldots, d_N\}$ and $d_i = \sum_j A_{ij}$; its normalized version is $\mathbf{L}_\mathrm{n} = \mathbf{D}^{-1/2}\mathbf{L}\mathbf{D}^{-1/2}$. Based on the eigendecomposition of the graph Laplacian $\mathbf{L} = \mathbf{U}\mathbf{\Lambda}\mathbf{U}^T$, where $\mathbf{U} \in \mathbb{R}^{N \times N}$ comprises of orthonormal eigenvectors and $\mathbf{\Lambda} = \mathrm{diag}\{\lambda_1, \ldots, \lambda_N\}$ is a diagonal matrix of eigenvalues, the graph Fourier transform is defined with eigenvectors of the graph Laplacian being the graph Fourier modes (harmonics) and the corresponding eigenvalues being the graph frequencies (Shuman et al., 2013). Assuming $\lambda_1 \leq \lambda_2 \leq \ldots \leq \lambda_N$, $\lambda_1$ corresponds to the lowest (zero) frequency and $\lambda_N$ corresponds to the highest frequency of the graph. For the case of normalized Laplacian $\mathbf{L}_\mathrm{n}$, all the graph frequencies lie in the range $[0, 2]$ (Shuman et al., 2013), with $\lambda_1 = 0$.

Let $\mathbf{x} \in \mathbb{R}^N$ be a single-channel input signal on the graph, then the graph Fourier transform and the inverse Fourier transform are defined as $\hat{\mathbf{x}} = \mathbf{U}^T\mathbf{x}$ and $\mathbf{x} = \mathbf{U}\hat{\mathbf{x}}$, respectively. Graph convolution of the input graph signal $\mathbf{x}$ with a filter $\mathbf{g}$ is

$$\mathbf{x} * \mathbf{g} := \mathbf{U}\left((\mathbf{U}^T\mathbf{x}) \odot (\mathbf{U}^T\mathbf{g})\right) = \mathbf{U}\hat{\mathbf{G}}\mathbf{U}^T\mathbf{x}, \tag{1}$$

where $\hat{\mathbf{G}} := \mathrm{diag}(\hat{\mathbf{g}}) = \mathrm{diag}\{\hat{g}_1, \ldots, \hat{g}_N\}$, and $\odot$ denotes element-wise multiplication.

### 3.2 Spectral Domain GNN Designs

The graph convolution given by (1) is used in GNNs to learn filters in the graph spectral domain. Spectral-GNN Bruna et al. (2014) learns all the filter coefficients, which requires full eigendecomposition of the Laplacian matrix and is very expensive. For computational efficiency, the filter coefficients $\hat{g}_1, \ldots, \hat{g}_N$ can be approximated via $K^{th}$ order polynomials of the graph frequencies $(K << N)$, i.e., $\hat{g}(\lambda_j) = \sum_{i=0}^{K} \theta_i \lambda_j^i$ with $\boldsymbol{\theta} \in \mathbb{R}^{K+1}$ being the (polynomial) filter coefficients. Then the graph convolution with such polynomial filter takes the form

$$\mathbf{x} * \mathbf{g} \approx \mathbf{U}\left(\sum_{i=0}^{K} \theta_i \mathbf{\Lambda}^i\right)\mathbf{U}^T\mathbf{x} = \sum_{i=0}^{K} \theta_i \mathbf{L}_\mathrm{n}^i \mathbf{x}. \tag{2}$$

Note that the polynomial filters allows us to write graph convolution in spatial domain via polynomial in the graph Laplacian. ChebNet (Defferrard et al., 2016) proposed to approximate the graph convolution via $K^{th}$ order Chebyshev polynomials. In GCN, (Kipf & Welling, 2017) simplified the graph convolution by assuming first order polynomial filter $(K = 1)$ with $\theta_0 = 2\theta$ and $\theta_1 = -\theta$, and thereby reducing the graph convolution to

$$\mathbf{x} * \mathbf{g} \approx \theta\left(2\mathbf{I} - \mathbf{L}_\mathrm{n}\right)\mathbf{x} = \theta\left(\mathbf{I} + \mathbf{D}^{-1/2}\mathbf{A}\mathbf{D}^{-1/2}\right)\mathbf{x}. \tag{3}$$

As a different interpretation, the above can also be viewed as a combination of two operations: (i) Feature aggregation via term $(\mathbf{I} + \mathbf{D}^{-1/2}\mathbf{A}\mathbf{D}^{-1/2})$ and (ii) Feature transformation via learnable parameter $\theta$. Note that the feature aggregation operation corresponds to low pass filtering since the spectral response of the spatial filter $2\mathbf{I} - \mathbf{L}_\mathrm{n}$ is $\hat{g}(\lambda) = 2 - \lambda$ which amplifies low-frequency signal $(\lambda \approx 0)$ and restrains high-frequency signal $(\lambda \approx 2)$. In its final design, for numerical stability, the feature aggregation operation in GCN is modified

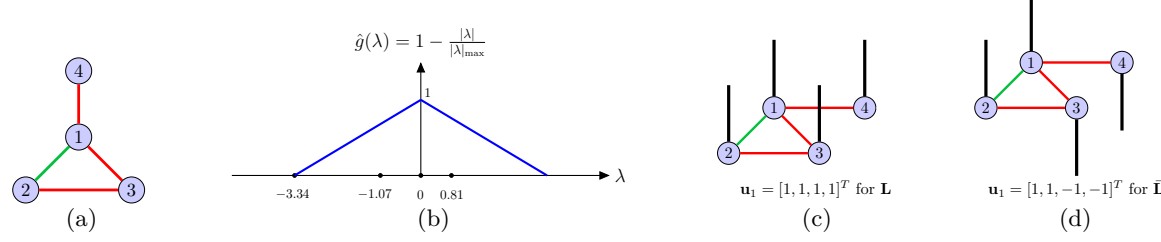

Figure 2: (a) A toy signed graph (Red color edges represent negative links and Green color edges represent positive links), (b) corresponding low-pass filter based on the unnormalized Laplacian, (c) Eigenvector corresponding to the lowest frequency based on the unsigned Laplacian, and (d) Eigenvector corresponding to the lowest frequency based on the signed Laplacian.

by adding self-loops for each node and as a result the modified graph convolution takes the form (Kipf & Welling, 2017; Wu et al., 2019)

$$\mathbf{x} * \mathbf{g} \approx \theta \ (\tilde{\mathbf{D}}^{-1/2} \tilde{\mathbf{A}} \tilde{\mathbf{D}}^{-1/2}) \ \mathbf{x}, \tag{4}$$

where $\tilde{\mathbf{A}} = \mathbf{A} + \mathbf{I}$ and $\tilde{\mathbf{D}} = \mathbf{D} + \mathbf{I}$. When generalized to multi-channel input $\mathbf{X}$, the output of the $\ell^{th}$ layer of the GCN reads

$$\mathbf{H}^{(\ell)} = \sigma(\mathbf{P} \ \mathbf{H}^{(\ell-1)} \ \mathbf{\Theta}^{(\ell)}), \qquad \mathbf{H}^{(0)} = \mathbf{X}, \tag{5}$$

where $\mathbf{P} = \tilde{\mathbf{D}}^{-1/2} \tilde{\mathbf{A}} \tilde{\mathbf{D}}^{-1/2}$ is the low-pass feature aggregation filter, $\mathbf{\Theta}^{(\ell)}$ is a learnable transformation matrix, and $\sigma$ is non-linearity such as ReLU.

### 3.3   Issues with Signed Graphs

In each GCN layer, the feature aggregation operation corresponds to low pass filtering with the filter being first order polynomial in $\mathbf{L}_n$. However for signed graphs, the inverse of the degree matrix $\mathbf{D}$ (or $\tilde{\mathbf{D}}$) becomes problematic since the degree values might be zero or negative values for some nodes and as a consequence the normalized Laplacian $\mathbf{L}_n$ is not well defined. Since the normalized Laplacian is not well defined, one is tempted to interpret the aggregation operator as a low pass filter based on unnormalized Laplacian $\mathbf{L}$. This again poses difficulty in frequency ordering as the eigenvalues of the Laplacian $\mathbf{L}$ can take negative values for signed graphs. The graph frequencies (Laplacian eigenvalues) are ordered based on the total variation (TV) of the corresponding eigenvectors on the graph (Sandryhaila & Moura, 2014; Ortega et al., 2018). TV quantifies global smoothness (or variation) of a graph signal. For unsigned graphs, the quadratic form $\mathbf{x}^T \mathbf{L} \mathbf{x}$ is often used as TV of signal $\mathbf{x}$ on the graph (Shuman et al., 2013). However for signed graphs, the quadratic form $\mathbf{x}^T \mathbf{L} \mathbf{x}$ may take negative values and thereby invalidating its use as TV. Another definition of TV of a graph signal $\mathbf{x}$ is (Singh et al., 2016) $\text{TV}(\mathbf{x}) = ||\mathbf{L}\mathbf{x}||_1$ and it can be shown that $\text{TV}(\mathbf{u}_i) > \text{TV}(\mathbf{u}_j)$, if $|\lambda_i| > |\lambda_j|$. Thus the eigenvalues with smaller absolute values act as low frequencies and vice-versa.

Although one can order the signed frequencies based on the absolute eigenvalues of the graph Laplacian $\mathbf{L}$, one needs new designs of low pass filters to be used as aggregation operator. For example, a low pass filter for the toy graph in Figure 2a with frequency response $\hat{g}(\lambda) = 1 - \frac{|\lambda|}{|\lambda|_{\max}}$ is shown in Figure 2b where $|\lambda|_{\max}$ is the maximum absolute frequency of the underlying graph. This filter design has certain drawbacks since it requires computation of $|\lambda|_{\max}$ and cannot be directly realized as a first order polynomial (in the graph Laplacian) in the spatial domain as the latter corresponds to a straight line. One can go for higher order filters with additional computational cost.

Moreover, the frequency interpretation also becomes ambiguous for signed graphs. Since the zero eigenvalue being the lowest frequency and the corresponding eigenvector being a constant signal on the graph (as shown in Figure 2c), it is intuitive only under similarity assumption (i.e. positive edges). For example, foes/enemies (users connected via negative links) having similar opinions (traits or features) suggests a high amount of variation and friends (users connected via positive links) having similar opinions constitutes small variation. However, this intuition is violated when using unsigned Laplacian: lowest frequency eigenvector values at nodes 1 and 3 connected via negative links in Figure 2c have the same values. Our proposed methods address these issues to design interpretable spectral domain signed GNNs.

# 4  Proposed Method

In this section we present our spectral domain analysis approach to graph neural networks for signed graphs. We then instantiate the approach to two specific network designs.

## 4.1  Spectral Domain Analysis of Signed Graphs

To address the issues mentioned in Section 3.3 while retaining the frequency interpretation of the aggregation operation, we turn to the signed graph signal processing (Dittrich & Matz, 2020). Instead of the standard graph Laplacian, we consider the signed Laplacian matrix (Kunegis et al., 2010; Dittrich & Matz, 2020) $\bar{\mathbf{L}} = \bar{\mathbf{D}} - \mathbf{A}$, where $\bar{\mathbf{D}} = \mathrm{diag}\{\bar{d}_1, \ldots, \bar{d}_N\}$ is a diagonal matrix with $\bar{d}_i = \sum_j |A_{ij}|$. In (Dittrich & Matz, 2020) the authors formalized the spectral domain analysis of the signals over signed graph via eigendecomposition of the signed Laplacian matrix with the eigenvalues being the signed graph frequencies and the corresponding eigenvectors being the signed graph harmonics.

More precisely, we consider the normalized signed Laplacian matrix

$$\bar{\mathbf{L}}_{\mathrm{n}} = \bar{\mathbf{D}}^{-1/2}\bar{\mathbf{L}}\bar{\mathbf{D}}^{-1/2} = \mathbf{I} - \bar{\mathbf{D}}^{-1/2}\mathbf{A}\bar{\mathbf{D}}^{-1/2}. \tag{6}$$

The eigenvalues of the normalized signed Laplacian lie in the range $[0, 2]$ with smaller eigenvalues corresponding to low frequencies, and vice-versa. This frequency ordering directly follows from using quadratic Laplacian form

$$\mathrm{TV}(\mathbf{x}) = \mathbf{x}^T \bar{\mathbf{L}}_{\mathrm{n}} \mathbf{x} \tag{7}$$

as the definition of TV on signed graphs. Using the eigenvectors of the signed Laplacian as graph harmonics provides natural frequency interpretation for signed graphs. The eigenvector corresponding to the lowest frequency of a signed graph is shown in Figure 2d. It can be seen that the nodes connected via negative links (foes) have opposite values and nodes connected via positive links (friends) have similar values thereby exhibiting small amount of variation; this phenomenon is intuitive for being a low frequency signal.

Our approach to signed graph neural networks naturally follows by redefining graph convolution based on the normalized signed Laplacian in (6). Building on this idea, we propose below two specific graph network designs for signed graphs: Spectral-SGCN-I and Spectral-SGCN-II. The former behaves like a low-pass filtering and can be viewed as a signed graph counterpart of the vanilla GCN (Kipf & Welling, 2017). The latter is able to retain high-frequency information and can be viewed as a signed graph counterpart of FAGCN (Bo et al., 2021).

## 4.2  Spectral-Signed-GCN-I

Our first network design, Spectral-SGCN-I, is similar to the vanilla GCN (Kipf & Welling, 2017). It can be viewed as a low-pass feature aggregation on the underlying signed graph followed by feature transformation. At each layer, the features are first aggregated via low-pass filter $\mathbf{P} = \tilde{\mathbf{D}}^{-1/2}\tilde{\mathbf{A}}\tilde{\mathbf{D}}^{-1/2}$ with $\tilde{\mathbf{A}} = \mathbf{A} + \mathbf{I}$ and $\tilde{\mathbf{D}} = \bar{\mathbf{D}} + \mathbf{I}$. It resembles (4) but uses the signed Laplacian. Note that, just like (4), we adopt the renormalization trick to improve numerical stability (Kipf & Welling, 2017).

In more details, the aggregated features in $\ell^{th}$ layer is $\bar{\mathbf{H}}^{(\ell)} = \mathbf{P}\mathbf{H}^{(\ell-1)}$. Let $\bar{\mathbf{H}}^{(\ell)} = [\bar{\mathbf{h}}_1^{(\ell)}, \bar{\mathbf{h}}_2^{(\ell)}, \ldots, \bar{\mathbf{h}}_N^{(\ell)}]^T$, then the aggregation can be written in the message passing form

$$\bar{\mathbf{h}}_i^{(\ell)} = \frac{1}{\bar{d}_i + 1}\mathbf{h}_i^{(\ell-1)} + \sum_{j \in \mathcal{N}_i^+}\frac{1}{\sqrt{(\bar{d}_i + 1)(\bar{d}_j + 1)}}\mathbf{h}_j^{(\ell-1)} - \sum_{k \in \mathcal{N}_i^-}\frac{1}{\sqrt{(\bar{d}_i + 1)(\bar{d}_k + 1)}}\mathbf{h}_k^{(\ell-1)}.$$

After aggregation, the features are transformed via a learnable parameter matrix along with non-linearity to give node representation output in $\ell^{th}$ layer as $\mathbf{H}^{(\ell)} = \sigma\left(\bar{\mathbf{H}}^{(\ell)}\,\mathbf{\Theta}^{(\ell)}\right)$.

The Spectral-SGCN-I aggregates only low frequency information via a low-pass aggregation filter as illustrated in Figure 3a. As it has been shown in (Wu et al., 2019) that removing nonlinearities and collapsing

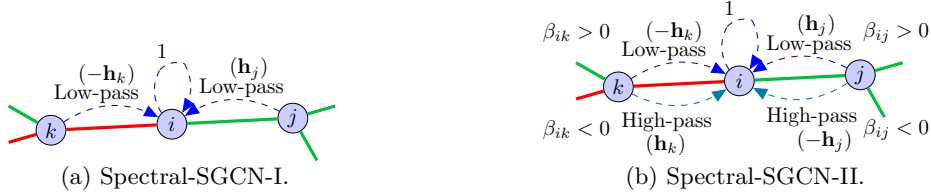

(a) Spectral-SGCN-I.  (b) Spectral-SGCN-II.

Figure 3: Spectral-SGCN-I propagates only low-pass information from its neighbors. Spectral-SGCN-II propagates low-pass as well as high-pass information from its neighbors via attention coefficients.

weight matrices between consecutive layers greatly simplifies the GCN complexity. Similarly, we propose spectral simplified signed graph convolution network (Spectral-S2GCN) such that the output of $\ell^{th}$ layer is $\mathbf{H}^{(\ell)} = \mathbf{P}^{\ell}\mathbf{X}\boldsymbol{\Theta}$, with $\boldsymbol{\Theta}$ being the only learnable parameter matrix.

### 4.3 Spectral-Signed-GCN-II

As has been noted in FAGCN (Bo et al., 2021), besides low frequency components, it is beneficial to incorporate the high frequency components during feature aggregation as well. Based on this idea and the frequency interpretation on signed graphs, we extend FAGCN to signed graphs by considering low as well as high frequency information. To this end, we use low pass filter $\mathbf{P}^{\text{Low}} = \mathbf{I} + \bar{\mathbf{D}}^{-1/2}\mathbf{A}\bar{\mathbf{D}}^{-1/2} = 2\mathbf{I} - \bar{\mathbf{L}}_n$ and high pass filter $\mathbf{P}^{\text{High}} = \mathbf{I} - \bar{\mathbf{D}}^{-1/2}\mathbf{A}\bar{\mathbf{D}}^{-1/2} = \bar{\mathbf{L}}_n$ along with attention to learn the proportion of low-frequency and high-frequency features to be propagated.

Let $\beta_{ij}^{\text{Low}}$ be the coefficient of attention aggregation for low frequency features from node $j$ to node $i$. Similarly, let $\beta_{ij}^{\text{High}}$ be the coefficient of attention aggregation for high frequency features from node $j$ to node $i$. Note $\beta_{ij}^{\text{Low}} = \beta_{ij}^{\text{High}} = 0$ if node $i$ is not connected to node $j$. For target node $i$, define low-pass attention matrix as $\mathbf{B}_i^{\text{Low}} = \text{diag}\{\beta_{i1}^{\text{Low}}, \ldots, \beta_{iN}^{\text{Low}}\}$ and high-pass attention matrix as $\mathbf{B}_i^{\text{High}} = \text{diag}\{\beta_{i1}^{\text{High}}, \ldots, \beta_{iN}^{\text{High}}\}$. Let $\mathbf{H}^{(\ell-1)} = [\mathbf{h}_1^{(\ell-1)}, \mathbf{h}_2^{(\ell-1)}, \ldots, \mathbf{h}_N^{(\ell-1)}]^T$ be the node embeddings at layer $\ell - 1$, then the $\ell^{th}$ GNN layer reads (assuming self-loops)

$$\mathbf{h}_i^{(\ell)} = \frac{1}{2}\left(\mathbf{P}^{\text{Low}}\mathbf{B}_i^{\text{Low}}\mathbf{H}^{(\ell-1)}\right)_i + \frac{1}{2}\left(\mathbf{P}^{\text{High}}\mathbf{B}_i^{\text{High}}\mathbf{H}^{(\ell-1)}\right)_i$$
$$= (\beta_{ii}^{\text{Low}} + \beta_{ii}^{\text{High}})\mathbf{h}_i^{(\ell-1)} + \sum_{j\in\mathcal{N}_i^+}\frac{(\beta_{ij}^{\text{Low}} - \beta_{ij}^{\text{High}})}{\sqrt{\bar{d}_i\bar{d}_j}}\mathbf{h}_j^{(\ell-1)} - \sum_{k\in\mathcal{N}_i^-}\frac{(\beta_{ik}^{\text{Low}} - \beta_{ik}^{\text{High}})}{\sqrt{\bar{d}_i\bar{d}_k}}\mathbf{h}_k^{(\ell-1)}.$$

The coefficient $\beta_{ij}^{\text{Low}} + \beta_{ij}^{\text{High}}$ acts as a scaling factor and can be set to be 1 for simplicity. Now denote $\beta_{ij}^{\text{Low}} - \beta_{ij}^{\text{High}} = \beta_{ij}$, then the above becomes

$$\mathbf{h}_i^{(\ell)} = \mathbf{h}_i^{(\ell-1)} + \sum_{j\in\mathcal{N}_i^+}\frac{\beta_{ij}}{\sqrt{\bar{d}_i\bar{d}_j}}\mathbf{h}_j^{(\ell-1)} - \sum_{k\in\mathcal{N}_i^-}\frac{\beta_{ik}}{\sqrt{\bar{d}_i\bar{d}_k}}\mathbf{h}_k^{(\ell-1)}. \tag{8}$$

When the attention coefficients are constant and equal to 1, the above reduces to Spectral-SGCN-I. In Spectral-SGCN-II, the attention coefficients are learned as $\beta_{ij} = \tanh\left(\mathbf{a}^T[\mathbf{h}_i, \mathbf{h}_j]\right)$ taking values in range $[-1, 1]$, where $\mathbf{a}$ is learnable linear parameter. When $\beta_{ij} = \beta_{ij}^{\text{Low}} - \beta_{ij}^{\text{High}} > 0$, the low-frequency information is propagated from node $j$ to node $i$ and when $\beta_{ij} < 0$, the high frequency information is propagated, as illustrated in Figure 3b. Before passing the given input features $\mathbf{X}$ to the first layer, they are first transformed to get $\mathbf{h}_i^{(0)} = \sigma\left(\boldsymbol{\Theta}_1\mathbf{x}_i\right)$ and after $L$ number of stacked layer, we get the final output embeddings as $\mathbf{h}_i = \boldsymbol{\Theta}_2\mathbf{h}_i^{(L)}$.

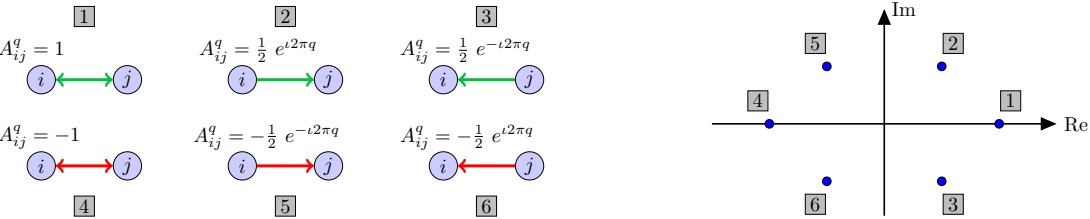

Figure 4: Illustration of different directed signed scenarios and signed directional phase locations on complex plane with $q = 0.125$.

## 5 Directed Signed Graphs

The methods proposed in Section 4 are limited to undirected signed graphs. For unsigned graphs, the magnetic Laplacian (Fanuel et al., 2017; 2018; Furutani et al., 2019) has been utilized to encode the edge directionality information. Recently, Zhang et al. (2021) used magnetic Laplacian for designing GNNs for directed (unsigned) graphs. In its original form, the magnetic Laplacian is defined as

$$\mathbf{L}^q = \mathbf{D} - \mathbf{A}^q = \mathbf{D} - \mathbf{A}_s \odot \mathbf{\Phi}^q, \tag{9}$$

where $\mathbf{A}_s$ is symmetric adjacency matrix with entries $\mathbf{A}_s(i,j) = \frac{1}{2}(\mathbf{A}(i,j) + \mathbf{A}(j,i))$, $\mathbf{D} = \text{diag}\{d_1, d_2, \ldots, d_N\}$ with $d_i = \sum_{j=1}^N \mathbf{A}_s(i,j)$. Moreover, $\mathbf{\Phi}^q$ is a Hermitian matrix with elements

$$\mathbf{\Phi}^q(i,j) = e^{\iota 2\pi q(\mathbf{A}(i,j) - \mathbf{A}(j,i))}, \tag{10}$$

where $\iota$ is an indeterminate satisfying $\iota^2 = -1$ and $q \in [0, 0.50)$ is the phase parameter. The normalized magnetic Laplacian is $\mathbf{L}_n^q = \mathbf{D}^{-1/2}\mathbf{L}^q\mathbf{D}^{-1/2}$.

It can be shown that $\mathbf{L}^q$ as well as $\mathbf{L}_n^q$ are positive semidefinite for the unsigned case and the eigenvalues of $\mathbf{L}_n^q$ lie in the interval $[0, 2]$. However, when the underlying graph is signed, the degree matrix can have zero diagonal entries and the normalized magnetic Laplacian is not well defined. Moreover, $\mathbf{L}^q$ becomes indefinite matrix for signed graphs. To handle these issues in directed signed graphs, we introduce *signed magnetic Laplacian*.

### 5.1 Signed Magnetic Laplacian

We define signed magnetic Laplacian as

$$\bar{\mathbf{L}}^q := \bar{\mathbf{D}} - \mathbf{A}^q = \bar{\mathbf{D}} - \mathbf{A}_s \odot \mathbf{\Phi}^q, \tag{11}$$

where $\mathbf{A}^q = \mathbf{A}_s \odot \mathbf{\Phi}^q$ contains the directional signed information via $\mathbf{\Phi}^q$ and the degree matrix is considered to be $\bar{\mathbf{D}} = \text{diag}\{\bar{d}_1, \bar{d}_2, \ldots, \bar{d}_N\}$ with $\bar{d}_i = \sum_{j=1}^N |\mathbf{A}_s(i,j)|$ representing the connection strength of node $i$. Note that the phase parameter $q \in [0, 0.25)$ for this signed directed settings (see Figure 4 for illustration of different scenarios). The normalized signed magnetic Laplacian is $\bar{\mathbf{L}}_n^q := \bar{\mathbf{D}}^{-1/2}\bar{\mathbf{L}}^q\bar{\mathbf{D}}^{-1/2}$. The eigendecomposition of $\bar{\mathbf{L}}_n^q$ can be used for spectral analysis of directed signed graphs and directed signed convolution operations can be defined as in Section 3.2. The spectrum of our signed magnetic Laplacians follow desirable properties as given by the following propositions (see Appendix for proofs).

**Proposition 1.** *The signed magnetic Laplacian $\bar{\mathbf{L}}^q$ and normalized signed magnetic Laplacian $\bar{\mathbf{L}}_n^q$ are positive semidefinite.*

**Proposition 2.** *The eigenvalues of the normalized signed magnetic Laplacian $\bar{\mathbf{L}}_n^q$ lie in $[0, 2]$.*

## 5.2 Directed Signed Graph Convolution Network

Based on signed magnetic Laplacian, similar to MagNet (Zhang et al., 2021), we propose Signed-MagNet. In its aggregation operation, Signed-MagNet aggregates features by performing low-pass filtering as

$$\bar{\mathbf{h}}_i^{(\ell)} = \frac{1}{\bar{d}_i + 1} \mathbf{h}_i^{(\ell-1)} + \sum_{j \in \mathcal{N}_i} \frac{\mathbf{A}^q(i,j)}{\sqrt{(\bar{d}_i + 1)(\bar{d}_j + 1)}} \mathbf{h}_j^{(\ell-1)}, \tag{12}$$

where $\mathcal{N}_i$ is the set of all the nodes connected to/from node $i$. Note that the latent embeddings in signed-MagNet are complex and at the last layer, we concatenate the real and imaginary parts. After aggregation operation in each layer, the features are transformed via a learnable complex matrix $\mathbf{\Theta}$.

**Remark 1.** *Note that we can also extend attention mechanism described in Section 4.3 to directed signed graphs in order to take low as well as high frequency information into account.*

# 6 Experiments

We evaluate our proposed methods for node classification and link sign predictions tasks on signed networks. We used Deep Graph Library (DGL) (Wang et al., 2019) for implementation of our methods. We also utilized PyTorch Geometric Signed Directed (He et al., 2022c) for implementing existing signed GNN baselines for node classification tasks. The code is available at `https://github.com/rahulsinghchandraul/Spectral_Signed_GNN`.

## 6.1 Datasets

We perform node classification task on three datasets: Wiki-Editor, Wiki-Election, and Wiki-RfA. Wiki-Editor is extracted from the UMDWikipedia dataset (Kumar et al., 2015). There is a positive edge between two users if their co-edits belong to the same categories and a negative edge represents the co-edits belonging to different categories. Each node is labeled as either benign or vandal. Wiki-RfA (West et al., 2014) and Wiki-Election (Leskovec et al., 2010) are datasets of editors of Wikipedia that request to become administrators. A request for adminship (RfA) is submitted, either by the candidate or by another community member and any Wikipedia member may give a supporting, neutral, or opposing vote. From these votes a signed network is built for each dataset, where a positive (resp. negative) edge indicates a supporting (resp. negative) vote by a user and the corresponding candidate. The label of each node in these networks is given by the output of the corresponding request: positive (resp. negative) if the editor is chosen (resp. rejected) to become an administrator. We use dataset extraction code provided by Mercado et al. (2019) [1].

For link sign prediction, we use three additional datasets: Bitcoin-Alpha, Bitcoin-OTC, and Slashdot [2]. Bitcoin-Alpha and Bitcoin-OTC (Kumar et al., 2016; 2018) are two exchanges in Bitcoins, where nodes represent Bitcoin users and edges represent the level of trust/distrust they have in other users. A positive edge implies trust, while a negative edge represents distrust (fraud). Slashdot dataset (Kunegis et al., 2009) is a network of interactions among users on Slashdot. Nodes represent users and edges represent friends (positive) or foes (negative). See Appendix C for more on the dataset statistics.

## 6.2 Results

We first present experiments on synthetic data generated by signed stochastic block model Cucuringu et al. (2019); He et al. (2022b) with different levels of imbalance. We simulate two clusters with intra-cluster edge probability of 0.02 having positive signs and inter-cluster probability of 0.01 having negative signs. Such a graph is a balanced graph. We then flip the inter-cluster as well as intra-cluster edge signs with different probabilities $q$ creating varying levels of imbalance. With only 2 node labels known per cluster for a total of 1000 nodes, the node classification performance is shown in Figure 5 (one hot vectors as input features and hidden dimension of 16). Binary cross-entropy loss based on the known labels is used as a loss function. Clearly, our proposed methods outperform SGCN (Derr et al., 2018) and SNEA (Li et al., 2020) significantly.

---

[1] https://github.com/melopeo/GL
[2] https://networks.skewed.de

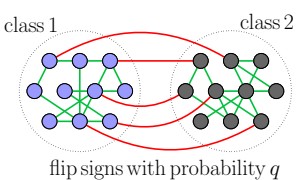
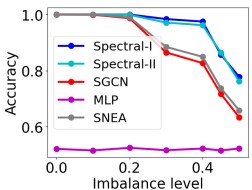

Figure 5: Performance with varying levels of imbalance.

Table 2: Test accuracies averaged over 20 different seeds for data shuffles and initializations.

| Dataset | WikiEditor | | | WikiElection | | | WikiRfA | | |
|---|---|---|---|---|---|---|---|---|---|
| Known Labels | 1 % | 2 % | 5 % | 1 % | 2 % | 5 % | 1 % | 2 % | 5 % |
| MLP | 72.3 ± 1.7 | 73.6 ± 1.1 | 74.8 ± 0.7 | 90.3 ± 0.9 | 93.7 ± 0.8 | 95.5 ± 0.6 | 91.5 ± 0.8 | 92.4 ± 1.2 | 93.8 ± 0.5 |
| GCN | 73.0 ± 0.8 | 73.3 ± 0.9 | 74.5 ± 1.0 | 88.9 ± 3.4 | 89.8 ± 3.7 | 89.9 ± 3.8 | 88.1 ± 2.2 | 89.4 ± 2.2 | 90.0 ± 2.8 |
| SAGE | 70.3 ± 0.8 | 72.8 ± 0.9 | 76.1 ± 0.5 | 87.2 ± 1.2 | 89.7 ± 1.2 | 93.3 ± 0.7 | 87.9 ± 0.9 | 89.9 ± 0.8 | 93.3 ± 0.6 |
| FAGCN | 75.4 ± 1.1 | 77.2 ± 0.9 | 78.8 ± 0.5 | 92.1 ± 1.4 | 93.3 ± 0.9 | 95.4 ± 0.7 | 91.5 ± 1.0 | 92.4 ± 0.8 | 94.1 ± 0.4 |
| SGCN | 79.4 ± 0.7 | 80.3 ± 0.9 | 82.1 ± 0.4 | 87.5 ± 1.2 | 89.9 ± 0.7 | 93.8 ± 0.4 | 87.9 ± 1.2 | 89.5 ± 1.7 | 91.3 ± 1.5 |
| SNEA | 79.5 ± 1.1 | 79.9 ± 1.0 | 81.1 ± 1.3 | 93.1 ± 1.5 | 94.1 ± 1.7 | 95.2 ± 1.7 | 93.0 ± 1.0 | 94.0 ± 1.1 | 94.4 ± 1.2 |
| SDGNN | 80.5 ± 1.3 | 81.1 ± 0.9 | 82.2 ± 0.7 | 95.8 ± 1.0 | 96.1 ± 0.6 | 96.7 ± 0.8 | 94.8 ± 1.1 | 95.0 ± 0.7 | 95.3 ± 0.4 |
| Spectral-SGCN-I | **81.1 ± 0.8** | 81.3 ± 0.8 | 82.8 ± 0.4 | 95.7 ± 1.0 | 97.1 ± 0.5 | 97.6 ± 0.4 | 94.4 ± 0.7 | 95.1 ± 0.5 | 95.7 ± 0.6 |
| Spectral-SGCN-II | 80.0 ± 2.2 | 80.9 ± 1.5 | 82.3 ± 1.4 | 96.7 ± 1.0 | 97.4 ± 0.3 | **98.0 ± 0.3** | 94.9 ± 0.4 | 95.6 ± 0.4 | 96.0 ± 0.2 |
| Spectral-S2GC | 81.1 ± 0.8 | 81.8 ± 0.7 | 82.7 ± 0.7 | 96.6 ± 0.9 | 97.3 ± 0.5 | 97.8 ± 0.2 | 94.3 ± 0.7 | 95.2 ± 0.4 | 95.9 ± 0.2 |
| Signed-MagNet | 81.0 ± 1.1 | **82.0 ± 0.8** | **83.0 ± 0.4** | **97.2 ± 0.6** | **97.6 ± 0.3** | 98.0 ± 0.2 | **96.0 ± 0.4** | **96.2 ± 0.3** | **96.4 ± 0.2** |

### 6.2.1 Node Classification

Next, we perform node classification task in a semi-supervised setting, i.e., we have access to the test data, but not the test labels, during training. For all the three datasets, we use three different ratios for training (known labels): 1%, 2%, 5% of the total nodes. Out of the remaining nodes, we use 90% for testing, and use rest of the nodes for validation. Since the features are not given for the nodes, we use truncated SVD of the symmetric adjacency matrix with dimension of 64 as input features. For comparison, we use traditional unsigned GNN designs namely GCN (Kipf & Welling, 2017), GraphSAGE (Hamilton et al., 2017), and FAGCN (Bo et al., 2021). For these methods, we do not consider the sign of the links, since the signed edge information is not applicable for these methods. We also use the state-of-the-art GNN designs based on balance theory including SGCN (Derr et al., 2018), SNEA (Li et al., 2020), and SDGNN (Huang et al., 2021) for comparison. For fair comparison, we use two layer networks with hidden dimension of 64 for all the GNN-based methods. Binary cross-entropy loss based on the known labels is used as a loss function. We use ReLU as the non-linearity function in between the layers. Adam is used as the optimizer along with $\ell_2$-regularization to avoid overfitting. We tune the learning rate and weight decay ($\ell_2$-regularization) hyperparameters over validation data using a grid search. For Signed-MagNet implementation, we fix $q = 0.125$ for all the experiments. Further details on implementation and hyperparameter tuning are provided in Appendix.

The classification results are summarized in Table 2. The experiments were run for 300 epochs and the results are averaged over 20 different random splits of training and test data. The average of best accuracies along with the standard deviation over 20 runs is reported. The best performing model for each dataset is in bold. We observe that our proposed signed GNNs consistently outperform the other methods in all the three datasets. For the two datasets Wiki-Election and Wiki-RfA, even traditional methods without signed information outperform SGCN.

### 6.2.2 Link Sign Prediction

Finally, we evaluate our methods for the task of link sign prediction that aims to predict the missing sign of a given edge. There exist three type of links in a signed graph: positive link, negative link, and no link. Denote this as a set $\mathcal{S} \in \{+, -, ?\}$, with "?" representing no link. Specifically, the training data contains a set of nodes $\mathcal{V}_t$ and a set of link triplets $\mathcal{T}$ consisting of triplets of the form $(u, v, s_{uv})$ with $u, v \in \mathcal{V}$ being node pairs and $s_{uv} \in \mathcal{S}$ denoting type of link between $u$ and $v$. The final embeddings (obtained from the GNN

Table 3: Link prediction results with Macro-F1 (left) and Micro-F1 (right) scores over 10 different runs.

| Method | Bitcoin-Alpha | | Bitcoin-OTC | | Slashdot | | WikiElection | | WikiEditor | | WikiRfA | |
|---|---|---|---|---|---|---|---|---|---|---|---|---|
| SiNE | 0.6790 | 0.9440 | 0.6832 | 0.9328 | 0.7238 | 0.8192 | 0.6940 | 0.7832 | 0.7858 | 0.8244 | 0.7263 | 0.8160 |
| SLF | 0.7475 | 0.9453 | 0.7483 | 0.9466 | **0.7943** | **0.8564** | 0.7616 | 0.8487 | 0.7521 | 0.8442 | **0.7881** | **0.8640** |
| SGCN | 0.6648 | 0.9184 | 0.7420 | 0.9012 | 0.7422 | 0.8260 | 0.7363 | 0.8360 | 0.8225 | 0.8548 | 0.7510 | 0.8430 |
| SNEA | 0.6796 | 0.9210 | 0.7580 | 0.9038 | 0.7431 | 0.8308 | 0.7388 | 0.8372 | 0.8290 | 0.8672 | 0.7542 | 0.8497 |
| SDGNN | 0.7412 | 0.9480 | 0.8020 | 0.9346 | 0.7820 | 0.8504 | 0.7550 | 0.8508 | 0.8529 | 0.8810 | 0.7832 | 0.8598 |
| Spectral-SGCN-I | 0.7518 | 0.9547 | 0.8371 | 0.9437 | 0.7705 | 0.8410 | 0.7466 | 0.8390 | 0.8560 | 0.9128 | 0.7740 | 0.8410 |
| Spectral-SGCN-II | 0.7630 | **0.9562** | 0.8356 | 0.9538 | 0.7724 | 0.8438 | 0.7645 | 0.8512 | **0.8825** | **0.9237** | 0.7735 | 0.8476 |
| Spectral-S2GCN | 0.7030 | 0.9418 | 0.7820 | 0.9284 | 0.7352 | 0.8194 | 0.7182 | 0.8390 | 0.8448 | 0.9083 | 0.7221 | 0.8278 |
| Singned-Magnet | **0.7880** | 0.9432 | **0.8548** | **0.9580** | 0.7781 | 0.8463 | **0.7818** | **0.8649** | 0.8652 | 0.9184 | 0.7692 | 0.8543 |

Table 4: Link prediction results with AUC scores over 10 different runs.

| Method | Bitcoin-Alpha | Bitcoin-OTC | Slashdot | WikiElection | WikiEditor | WikiRfA |
|---|---|---|---|---|---|---|
| SiNE | $0.8351 \pm 0.0126$ | $0.8575 \pm 0.0053$ | $0.8108 \pm 0.0021$ | $0.8040 \pm 0.0072$ | $0.8631 \pm 0.0044$ | $0.7963 \pm 0.0260$ |
| SLF | $0.8438 \pm 0.0151$ | $0.8670 \pm 0.0052$ | $\mathbf{0.8846 \pm 0.0040}$ | $0.8803 \pm 0.0025$ | $0.9090 \pm 0.0027$ | $\underline{0.8709 \pm 0.0012}$ |
| SGCN | $0.8420 \pm 0.0147$ | $0.8780 \pm 0.0103$ | $0.8543 \pm 0.0064$ | $0.8516 \pm 0.0030$ | $0.9068 \pm 0.0040$ | $0.8361 \pm 0.0078$ |
| SNEA | $0.8453 \pm 0.0062$ | $0.8792 \pm 0.0028$ | $0.8621 \pm 0.0154$ | $0.8412 \pm 0.0083$ | $0.9278 \pm 0.0062$ | $0.8259 \pm 0.0036$ |
| SDGNN | $0.9008 \pm 0.0081$ | $0.9128 \pm 0.0073$ | $\underline{0.8734 \pm 0.0187}$ | $0.8763 \pm 0.0134$ | $0.9430 \pm 0.0126$ | $\mathbf{0.8870 \pm 0.0048}$ |
| Spectral-SGCN-I | $0.9005 \pm 0.0345$ | $0.9079 \pm 0.0176$ | $0.8345 \pm 0.0121$ | $0.8559 \pm 0.0316$ | $0.9447 \pm 0.0194$ | $0.8228 \pm 0.0333$ |
| Spectral-SGCN-II | $\underline{0.9146 \pm 0.0066}$ | $\underline{0.9309 \pm 0.0044}$ | $0.8677 \pm 0.0090$ | $\underline{0.8840 \pm 0.0018}$ | $\mathbf{0.9818 \pm 0.0047}$ | $0.8420 \pm 0.0026$ |
| Spectral-S2GCN | $0.8670 \pm 0.0176$ | $0.8936 \pm 0.0095$ | $0.8273 \pm 0.0291$ | $0.8149 \pm 0.0315$ | $0.9375 \pm 0.0121$ | $0.8242 \pm 0.0189$ |
| Signed-MagNet | $\mathbf{0.9227 \pm 0.0097}$ | $\mathbf{0.9410 \pm 0.0076}$ | $0.8615 \pm 0.0074$ | $\mathbf{0.8881 \pm 0.0026}$ | $\underline{0.9567 \pm 0.0144}$ | $0.8612 \pm 0.0032$ |

model) of the two nodes $u$ and $v$ are concatenated together $[\mathbf{h}_u, \mathbf{h}_v]$ as the set of features for the edge and then fed to a three-class MLP classifier. The models are trained using the labeled edges from the training data. Let one-hot encoded vector of link type $s_{uv}$ be $\mathbf{s}_{uv} \in \{0,1\}^{|\mathcal{S}|}$. We use multi-class (three) cross entropy loss:

$$\mathcal{L}(\mathbf{\Theta}, \mathbf{W}) = -\frac{1}{|\mathcal{T}|} \sum_{(u,v,s_{uv}) \in \mathcal{T}} \sum_{c=1}^{|\mathcal{S}|} \mathbf{s}_{uv}(c) \log(\hat{\mathbf{s}}_{uv}(c)),$$

where $\hat{\mathbf{s}}_{uv}(c)$ is the predicted probability for class $c$ via softmax function. In the above loss function, $\mathbf{\Theta}$ denotes the set of GNN parameters and $\mathbf{W}$ denotes the parameters of MLP classifier. We use 80% of the links for training and rest 20% for testing. We use twice the number of training links as "no links" obtained using negative sampling.

We use SiNE (Wang et al., 2017), SLF (Xu et al., 2019), SGCN (Derr et al., 2018), SNEA (Li et al., 2020), and SDGNN (Huang et al., 2021) as baselines for comparison on link sign prediction tasks. We use a two layer GNN model along with a single hidden layer MLP classifier. Truncated SVD of the symmetric adjacency matrix with dimension of 30 is used as input features. We utilize Marco-F1 and Micro-F1 scores for evaluation, since the positive and negative links are unbalanced. The comparison results in terms of F1 scores are listed in Table 3. Moreover, the results in terms of Area Under the receiver operating characteristic Curve (AUC) scores in Table 4. The results reported in these tables are average scores over 10 different runs. We observe that our methods outperform the baselines for all the dataset except for Slashdot and WikiRfA. Note that all the numbers for other algorithms in the table are obtained by running the official codes provided by the respective authors. These could be slightly different from the numbers reported in those papers that may use filtered/truncated versions of datasets.

# 7 Conclusion

In this paper, we proposed a new framework for GNN design for signed graphs based on spectral domain analysis over signed graphs, as opposed to existing balance theory based GNN methods. We also introduced signed magnetic Laplacian for handling directed signed graphs. We evaluated our methods for node classification as well as link sign prediction tasks on signed graphs and achieved state-of-the-art performance.

**Limitations and Ethical Considerations:** One limitation of our method is that it does not consider one scenario in directed signed graphs when there exist two opposite (directed) edges with different signs between a pair of nodes. All of our data is publicly available for research purposes and does not contain personally identifiable information or offensive content. The methods presented here has no greater or lesser impact on society than other graph neural network algorithms.

## Acknowledgements

We would like to thank the anonymous reviewers for their helpful feedback on earlier versions of this text. This work was supported by NSF under grant 1942523.

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

Table 5: Notations used in this paper.

| Notations | Description |
|---|---|
| $\mathbb{R}^n$ | n-dimensional Euclidean space |
| $\mathbb{C}^n$ | n-dimensional complex coordinate space |
| $x, \mathbf{x}, \mathbf{X}$ | Scalar, vector, and matrix |
| $\mathbf{X}^T$ | Matrix transpose |
| $\mathbf{X}^\dagger$ | Matrix conjugate transpose |
| $N$ | Number of nodes in the graph |
| $\mathcal{N}_i$ | Set of nodes neighboring node $i$ |
| $\mathcal{N}_i^+$ | Set of neighbors connected to node $i$ via positive edges |
| $\mathcal{N}_i^-$ | Set of neighbors connected to node $i$ via negative edges |
| $\mathbf{A}$ | The adjacency matrix |
| $\mathbf{A}_s$ | The symmetric adjacency matrix |
| $\mathbf{D}$ | The degree matrix |
| $\mathbf{L}$ | The Laplacian matrix |
| $\mathbf{L}_\mathrm{n}$ | The normalized Laplacian matrix |
| $\bar{\mathbf{L}}$ | The signed Laplacian matrix |
| $\bar{\mathbf{L}}_\mathrm{n}$ | The normalized signed Laplacian matrix |
| $\mathbf{L}^q$ | The magnetic Laplacian matrix |
| $\mathbf{L}_\mathrm{n}^q$ | The normalized magnetic Laplacian matrix |
| $\bar{\mathbf{L}}^q$ | The signed magnetic Laplacian matrix |
| $\bar{\mathbf{L}}_\mathrm{n}^q$ | The normalized signed magnetic Laplacian matrix |
| $\mathbf{I}$ | Identity matrix |
| $\mathbf{x} * \mathbf{g}$ | Convolution of $\mathbf{x}$ and $\mathbf{g}$ |
| $\lambda$ | Graph frequency |
| $\mathbf{P}$ | A graph filter |
| $\boldsymbol{\Theta}^{(\ell)}$ | Learnable transformation matrix in the $\ell^{th}$ GNN layer |
| $\mathbf{h}_i^{(\ell)}$ | Latent feature vector of node $i$ in the $\ell^{th}$ GNN layer |
| $\mathbf{H}^{(\ell)}$ | All the latent feature vectors (each row corresponding to a node) in the $\ell^{th}$ GNN layer |
| $q$ | Phase parameter |
| $\sigma$ | A non-linear function |
| $\beta_{ij}$ | Attention coefficient from node $j$ to node $i$ |

## A  Proof of Proposition 1

*Proof.* Let $\mathbf{X}^\dagger$ be the conjugate transpose of $\mathbf{X}$ and let $\mathbf{x} \in \mathbb{C}^N$. It is easy to see that $\bar{\mathbf{L}}^q$ is a Hermitian matrix and therefore, the imaginary part $Im(\mathbf{x}^\dagger \bar{\mathbf{L}}^q \mathbf{x}) = 0$. Denoting $\boldsymbol{\Phi}^q(m, n) = e^{\iota \boldsymbol{\Theta}^q(m,n)}$, where $\boldsymbol{\Theta}^q(m, n) = 2\pi q(\mathbf{A}(m, n) - \mathbf{A}(n, m))$. The real part

$$
\begin{aligned}
Re(\mathbf{x}^\dagger \bar{\mathbf{L}}^q \mathbf{x}) &= \sum_{n=1}^{N} \bar{\mathbf{D}}(n, n)\mathbf{x}(n)\mathbf{x}^*(n) - \sum_{m,n=1}^{N} \mathbf{A}_s(m, n)\mathbf{x}(m)\mathbf{x}^*(n)\cos(\boldsymbol{\Theta}^q(m, n)) \\
&= \sum_{m,n=1}^{N} |\mathbf{A}_s(m, n)||\mathbf{x}(m)|^2 - \sum_{m,n=1}^{N} \mathbf{A}_s(m, n)\mathbf{x}(m)\mathbf{x}^*(n)\cos(\boldsymbol{\Theta}^q(m, n)) \\
&\geq \frac{1}{2}\sum_{m,n=1}^{N} |\mathbf{A}_s(m, n)||\mathbf{x}(m)|^2 + \frac{1}{2}\sum_{n,m=1}^{N} |\mathbf{A}_s(n, m)||\mathbf{x}(n)|^2 \\
&\quad - \sum_{m,n=1}^{N} |\mathbf{A}_s(m, n)\mathbf{x}(m)\mathbf{x}^*(n)\cos(\boldsymbol{\Theta}^q(m, n))| \\
&= \frac{1}{2}\sum_{m,n=1}^{N} |\mathbf{A}_s(m, n)|\left(|\mathbf{x}(m)|^2 + |\mathbf{x}(n)|^2 - 2|\mathbf{x}(m)\mathbf{x}^*(n)\cos(\boldsymbol{\Theta}^q(m, n))|\right) \\
&\geq \frac{1}{2}\sum_{m,n=1}^{N} |\mathbf{A}_s(m, n)|\left(|\mathbf{x}(m)|^2 + |\mathbf{x}(n)|^2 - 2|\mathbf{x}(m)||\mathbf{x}(n)|\right) \\
&= \frac{1}{2}\sum_{m,n=1}^{N} |\mathbf{A}_s(m, n)|\left(|\mathbf{x}(m)| - |\mathbf{x}(n)|\right)^2 \\
&\geq 0.
\end{aligned}
$$

Letting $\mathbf{y} = \bar{\mathbf{D}}^{-1/2}\mathbf{x}$ and by definition of $\bar{\mathbf{L}}_n^q$, we have

$$\mathbf{x}^\dagger \bar{\mathbf{L}}_n^q \mathbf{x} = \mathbf{x}^\dagger \bar{\mathbf{D}}^{-1/2} \bar{\mathbf{L}}^q \bar{\mathbf{D}}^{1/2} \mathbf{x} = \mathbf{y}^\dagger \bar{\mathbf{L}}^q \mathbf{y} \geq 0.$$

$\square$

## B  Proof of Proposition 2

*Proof.* Since $\bar{\mathbf{L}}_n^q$ is positive semidefinite due to Proposition 1, we just show that the largest eigenvalue $\lambda_N \leq 2$. We know that the eigenvalue with largest absolute value is

$$\lambda_N = \max_{\mathbf{x} \neq 0} \frac{\mathbf{x}^\dagger \bar{\mathbf{L}}_n^q \mathbf{x}}{\mathbf{x}^\dagger \mathbf{x}}.$$

Letting $\mathbf{y} = \bar{\mathbf{D}}^{-1/2}\mathbf{x}$, we have

$$\lambda_N = \max_{\mathbf{x} \neq 0} \frac{\mathbf{x}^\dagger \bar{\mathbf{D}}^{-1/2} \bar{\mathbf{L}}^q \bar{\mathbf{D}}^{1/2} \mathbf{x}}{\mathbf{x}^\dagger \mathbf{x}} = \max_{\mathbf{y} \neq 0} \frac{\mathbf{y}^\dagger \bar{\mathbf{L}}^q \mathbf{y}}{\mathbf{y}^\dagger \bar{\mathbf{D}} \mathbf{y}}.$$

Since the numerator in the above

$$\begin{aligned}
\mathbf{y}^\dagger \bar{\mathbf{L}}^q \mathbf{y} &= \sum_{m,n=1}^{N} |\mathbf{A}_s(m,n)||\mathbf{y}(m)|^2 - \sum_{m,n=1}^{N} \mathbf{A}_s(m,n)\mathbf{y}(m)\mathbf{y}^*(n)\cos(\mathbf{\Theta}^q(m,n)) \\
&= \frac{1}{2}\sum_{m,n=1}^{N} |\mathbf{A}_s(m,n)||\mathbf{y}(m)|^2 + \frac{1}{2}\sum_{m,n=1}^{N} |\mathbf{A}_s(m,n)||\mathbf{y}(n)|^2 \\
&\quad - \sum_{m,n=1}^{N} \mathbf{A}_s(m,n)\mathbf{y}(m)\mathbf{y}^*(n)\cos(\mathbf{\Theta}^q(m,n)) \\
&\leq \frac{1}{2}\sum_{m,n=1}^{N} |\mathbf{A}_s(m,n)|\left(|\mathbf{y}(m)| + |\mathbf{y}(n)|\right)^2 \\
&\leq \sum_{m,n=1}^{N} |\mathbf{A}_s(m,n)|\left(|\mathbf{y}(m)|^2 + |\mathbf{y}(n)|^2\right) \\
&= 2\sum_{m,n=1}^{N} |\mathbf{A}_s(m,n)|\,|\mathbf{y}(m)|^2 \\
&= 2\sum_{m=1}^{N} \left(\sum_{n=1}^{N} |\mathbf{A}_s(m,n)|\right)|\mathbf{y}(m)|^2 \\
&= 2\sum_{m=1}^{N} \bar{\mathbf{D}}(m,m)\,|\mathbf{y}(m)|^2 \\
&= 2\mathbf{y}^\dagger \bar{\mathbf{D}}\mathbf{y},
\end{aligned}$$

and thus, $\lambda_N \leq 2$. $\square$

## C  Datasets

The statistics of the real-world datasets used in our experiments are summarized in Table 6. Moreover, the histogram of the spectrum of unsigned Laplacian is depicted in Figure 6, from which the presence of negative eigenvalues is evident. This Laplacian is not normalizable due to presence of zero degree values. Moreover, we plot the spectrum of the normalized signed Laplacian in Figure 7.

Table 6: Signed graph datasets.

| | Wiki-Editor | Wiki-Elections | Wiki-RfA | Bitcoin-Alpha | Bitcoin-OTC | Slashdot |
|---|---|---|---|---|---|---|
| #Nodes | 21535 | 7194 | 11381 | 3783 | 5881 | 79120 |
| #Classes | 2 | 2 | 2 | - | - | - |
| #Positive Links | 269251 | 81862 | 139345 | 22650 | 32029 | 392179 |
| #Negative Links | 79004 | 22497 | 39433 | 1563 | 3563 | 123218 |
| %Positive Links | 77.31 % | 78.44 % | 77.94 % | 93.54 % | 89.99 % | 76.09 % |
| Eigenvalue range ($\mathbf{L}$) | [-422.56, 793.33] | [-115.75,646.22] | [-154.91, 708.21] | [-56.14,504.03] | [-63.90,782.02] | [-660.07, 2473] |
| %Negative Eigenvalues ($\mathbf{L}$) | 26.70 % | 24.31 % | 23.45 % | 5.77 % | 10.36 % | 21.98 % |

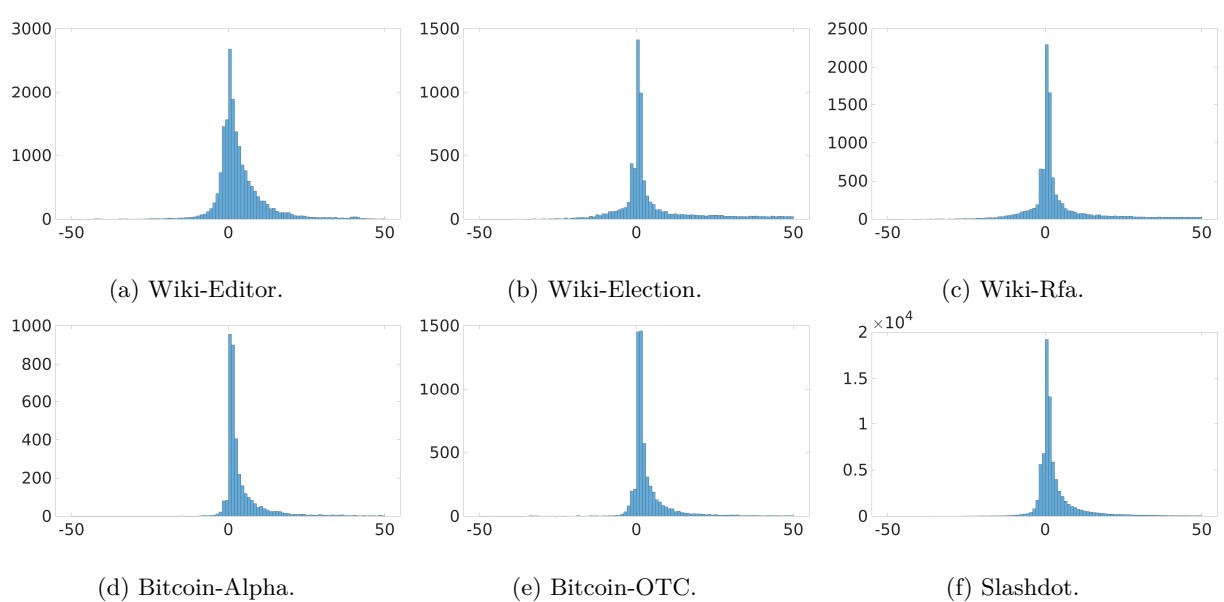

(a) Wiki-Editor.    (b) Wiki-Election.    (c) Wiki-Rfa.

(d) Bitcoin-Alpha.    (e) Bitcoin-OTC.    (f) Slashdot.

Figure 6: Histogram of the Laplacian ($\mathbf{L}$) eigenvalues for different datasets.

# D    Further Implementation Details

All the experiments were run on Intel Core i9-9900 machine equipped with NVIDIA GeForce RTX 2080 Ti GPU. We use two layer networks with hidden dimension of 64 for all the GNN-based methods (a standard practice in unsigned GNN literature). ReLU nonlinearity was used in all the experiments. For the implementation of Signed-Magnet, we used ReLU non-linearity for real and complex parts separately. The only hyperparameters to tune are learning rate and regularization (weight decay) coefficient. For all of our methods we use feature dropout with a rate of 0.5. For Spectral-SGCN-II, we use attention and feature dropout with dropout rate of 0.5. We tune the learning rate with different values (on log scale) in the range $[1e^{-3}, 1e^{-1}]$ and regularization rate in the range $[1e^{-6}, 1e^{-3}]$. For node classification task, the hyperparameters were tuned based on the validation accuracy with 1% known training labels for each dataset.

**Complexity Analysis:** Although the time complexity of the existing signed GNN designs (balance theory based) and our methods is $\mathcal{O}(|E|)$ for sparse graphs or $\mathcal{O}(|V|^2)$ in worst-case, the number of parameters per layer for these methods are different. For example, SDGNN utilizes four different encoders for capturing different directionality scenarios. In contrast, our signed spectral methods employ a single (transformation) encoder in each layer. We provide training times per epoch (averaged over 300 epochs) for different signed GNN methods in Figure 8.

# E    Clustering of Directed Signed Graphs

We present experiments on synthetic data generated by directed signed stochastic block model with different levels of imbalance. In particular, we simulate two clusters (classes) with directed intra-cluster edge probability of 0.05 having positive signs and directed inter-cluster edge probability of 0.05 having negative signs. Such a

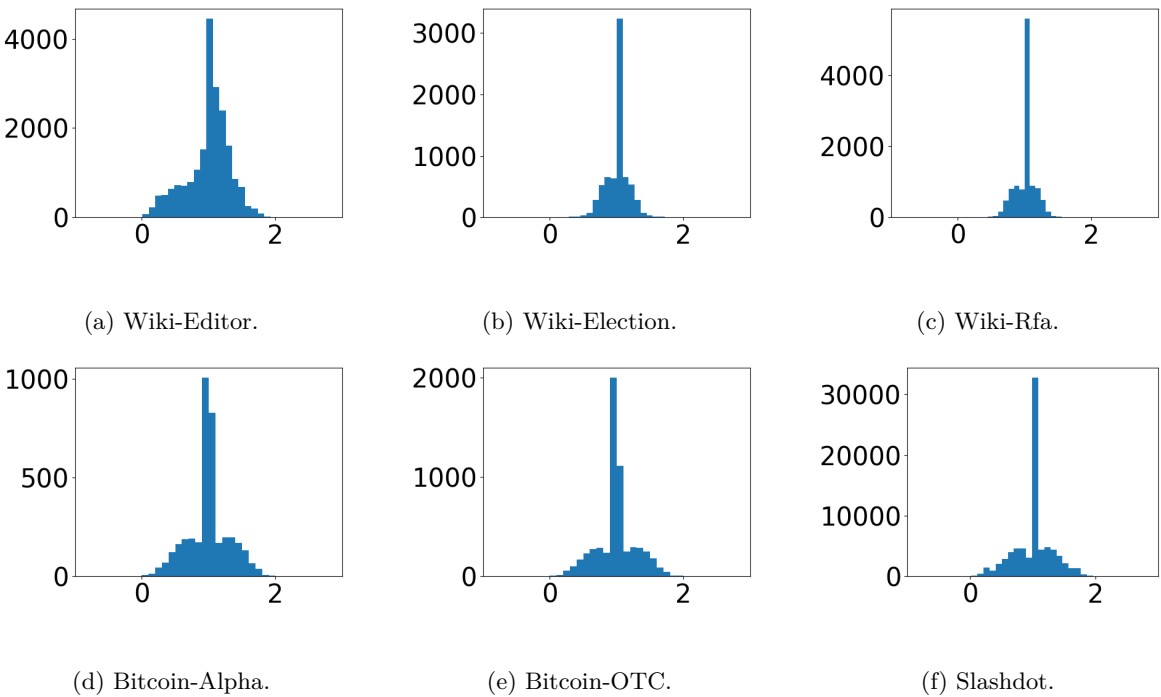

Figure 7: Histogram of the normalized signed Laplacian ($\bar{\mathbf{L}}_{\mathbf{n}}$) eigenvalues for different datasets.

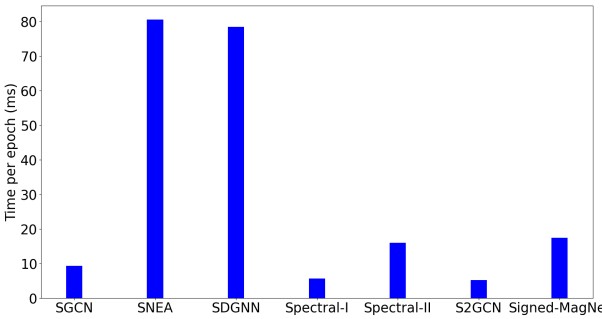

Figure 8: Time taken per epoch for different methods (on WikiRfA dataset for node classification task with batch size = 64, number of GNN layers = 2, hidden dimension = 64).

graph is a balanced graph. We then flip the inter-cluster as well as intra-cluster edge signs with different probabilities creating varying levels of imbalance.

The nodes can be clustered based on the first (complex) eigenvector of the signed magnetic Laplacian corresponding to the eigenvalue with smallest absolute value plotted on the complex plane. Figure 9 shows clustering results on directed signed graphs with varying levels of imbalance.

# F  Connections to SGCN

SGCN (Derr et al., 2018) in its design consider balanced and unbalanced node sets based on balance theory in feature aggregation process. The balanced node set for a target node $i$ is the set of nodes that have even number of negative links along a path connecting to $i$. An $\ell$-hop balanced set of nodes for target node $i$ is

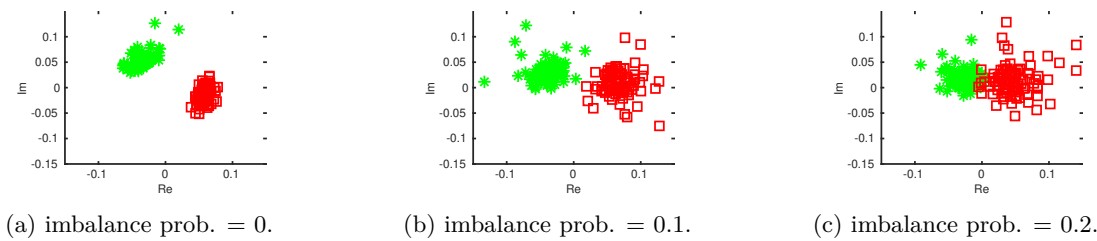

(a) imbalance prob. = 0.    (b) imbalance prob. = 0.1.    (c) imbalance prob. = 0.2.

Figure 9: Clustering results with 100 nodes per cluster for directed signed stochastic block model.

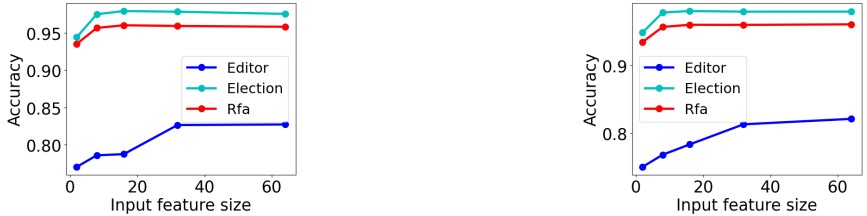

Figure 10: Spectral-SGCN-I (left) and Spectral-SGCN-II (right) performance with varying dimensions of input features (5% known labels).

denoted as $B_i(\ell)$ and unbalanced set of nodes as $U_i(\ell)$. For example graph in Figure 2, $B_1(1) = \{2\}$ and $U_1(1) = \{3, 4\}$.

The node representations for these balanced and unbalanced sets are treated separately in feature aggregation process and are concatenated together to form final node embeddings. In the $\ell^{th}$ layer of the model, it reads

$$\mathbf{h}_i^{B(\ell)} = \sigma\left(\Theta^{B(\ell)}\left[\sum_{j\in\mathcal{N}_i^+}\frac{\mathbf{h}_j^{B(\ell-1)}}{|\mathcal{N}_i^+|}, \sum_{k\in\mathcal{N}_i^-}\frac{\mathbf{h}_j^{U(\ell-1)}}{|\mathcal{N}_i^-|}, \mathbf{h}_i^{B(\ell-1)}\right]\right)$$

$$\mathbf{h}_i^{U(\ell)} = \sigma\left(\Theta^{U(\ell)}\left[\sum_{j\in\mathcal{N}_i^+}\frac{\mathbf{h}_j^{U(\ell-1)}}{|\mathcal{N}_i^+|}, \sum_{k\in\mathcal{N}_i^-}\frac{\mathbf{h}_j^{B(\ell-1)}}{|\mathcal{N}_i^-|}, \mathbf{h}_i^{U(\ell-1)}\right]\right),$$

where $\Theta^{B(\ell)}$ and $\Theta^{U(\ell)}$ are linear transformation parameters for balanced and unbalanced paths, respectively and the node representation at $\ell^{th}$ layer is the concatenation of the two embeddings $\mathbf{h}^{(\ell)} = [\mathbf{h}_i^{B(\ell)}, \mathbf{h}_i^{U(\ell)}]$.

Instead of treating positive and negative neighbors separately, in our architecture of Spectral-SGCN-I we are aggregating them weighted by their signs and corresponding (absolute) degrees as can be seen from (12).

## G   Effect of Input Feature Size

We also perform the study on the classification accuracy with varying number of input feature sizes. Figure 10 shows the performance of Spectral-SGCN-II with respect to varying number of input features. As expected, the performance improves with increase in the dimension of input features.

