# OpenReview forum: "Signed Graph Neural Networks: A Frequency Perspective"
_TMLR — Accepted by TMLR_

### Review · Reviewer_Ua9S · 2022-10-21

**Summary Of Contributions:**

The paper proposes a new method for handling graphs with signed connections, by analysing the spectral representation of the graph.  The method is accompanied by some theoretical analysis.  The method is then benchmarked fairly thoroughly against other methods, and appears to perform favourably.

**Audience:**

Yes

**Broader Impact Concerns:**

No concerns.

**Claims And Evidence:**

Yes

**Requested Changes:**

I consider all of these changes "critical", since they are all easy-to-change typographical things that would help the reader.

(i)  Insert a short additional "Problem Definition" section before Section 2.  This will take some of the notation from the first paragraph of what is currently Section 2.  This section should concretely define the problem structure:  what are the inputs and outputs (and their spaces), what is the loss function (or at least the type of loss function), what are the parameters, what constituted a "good" outcome, what does a typical application  look like etc.  There should be little-to-no math in this section, other than defining terms.  Currently, you introduce some preliminaries, and then dive straight in to very low-level details, and never really "pop back up".  It is also reasonable as the final paragraph in this section to (re-)outline how and what your method will overcome.

(ii)  A banner diagram at the top of page 2 that visually describes the problem and solution.  I think Figure 1 is a good starting point for this, but I would like to see it enriched with an illustration of the method (if possible).  This diagram should prime the reader for the material that is subsequently presented.

(iii)  I would like a table of notation added as the first section in the supplement, since the paper is very notationally heavy.

(iv)  Add introduction of core terminology and ideas.

**Strengths And Weaknesses:**

I will preface my review by stating that I am not really qualified or experienced enough to review the technical content in this paper.  I have **no** experience with graph methods, spectral methods, Laplacians or the datasets and problem domains tackled.  Therefore, my review will be somewhat high-level.

## Strengths
The paper is fairly well written and appears to be thoroughly benchmarked on a number of widely used examples.  From what I can gather, it also seems to be a timely and novel method that is well-placed amongst several recent other publications.

## Weaknesses
My main complaints about the paper are centred on the presentation.

(1)  The authors have clearly done a lot of good work.  However, the paper is very intense, and requires the reader to already be an expert in graph and spectral methods to get **anything** out of the paper.  Basic terms and concepts are not really defined anywhere (e.g. how does a graph have a frequency...?).  I think this is okay... but will severely limit the reach and impact of the paper.  The paper would benefit from introducing more slowly some of the basic terminology and ideas.  (Also see (i) below.)

(2)  The review of existing work feels like more of a hat-tipping exercise than actually introducing the reader to the landscape of the field.  The weaknesses of existing methods are curtly stated, and not really fleshed out at all.

(3)  The paper can easily be reduced in length to make space for my recommendations.  Some of the information on the datasets can be cut to the supplement;  paragraphs two and three in the intro can be combined + shortened + thinned to the relevant work section;  I believe much of Section two could be made shorter and more direct.

## Minor Weaknesses / Typographical Comments
(a)  Figures should be floated to the top of pages (cf. Figure 1, 3, 4, Table 1).

(b)  \eqref should be used for referencing equations, i.e. "equation \ref{X}" -> "\eqref{X}".

(c)  Footnotes should be avoided where possible.

(d)  What is $\mathcal{V}$ in the definition of $\mathcal{G}$?  It doesn't seem to be used anywhere.


## Summary
Overall, I think the submission itself is of publication quality, though I cannot reasonably comment on the "science".  There is a good amount of work to be done to make the paper more accessible (see "Requested Changes").

---

> ### Author Response · Authors · 2022-12-28
> **Response to Reviewer Ua9S**
>
> Thank you very much for the comments and suggestions. We have updated the manuscript accordingly. The major changes we have made are as follows.
> 1. We have revised the Introduction, Related Work, and Preliminaries sections significantly.
> 2. We have included Problem Definition section in the main text and notations table in Appendix.
> 3. We have included the Laplacian spectrum of all the real-world datasets in Appendix.
>
> **(1) The paper is very intense, and requires the reader to already be an expert in graph and spectral methods to get anything out of the paper. Basic terms and concepts are not really defined anywhere (e.g. how does a graph have a frequency...?). I think this is okay... but will severely limit the reach and impact of the paper. The paper would benefit from introducing more slowly some of the basic terminology and ideas.**
>
> We have modified the Preliminaries section significantly for better clarity. We have added a new subsection "Graph Fourier Transform and Spectral Filtering" in Preliminaries section explaining key terms and concepts of spectral filtering utilized in spectral GNN designs. We have also added a separate Problem definition Section for better readability.
>
> **(2) The review of existing work feels like more of a hat-tipping exercise than actually introducing the reader to the landscape of the field. The weaknesses of existing methods are curtly stated, and not really fleshed out at all.**
>
> Thanks for pointing this out. We have included more descriptions of the existing methods in the Related Work section.
>
> **(3) The paper can easily be reduced in length to make space for my recommendations. Some of the information on the datasets can be cut to the supplement; paragraphs two and three in the intro can be combined + shortened + thinned to the relevant work section; I believe much of Section two could be made shorter and more direct**
>
> We have shortened dataset description and moved the dataset statistics Table to Appendix D. We have modified the Introduction and extended the Related Work section. We have also modified the Preliminaries section significantly for better clarity and explanations.
>
> **(i) Insert a short additional "Problem Definition" section before Section 2. This will take some of the notation from the first paragraph of what is currently Section 2. This section should concretely define the problem structure: what are the inputs and outputs (and their spaces), what is the loss function (or at least the type of loss function), what are the parameters, what constituted a "good" outcome, what does a typical application look like etc. There should be little-to-no math in this section, other than defining terms. Currently, you introduce some preliminaries, and then dive straight in to very low-level details, and never really "pop back up". It is also reasonable as the final paragraph in this section to (re-)outline how and what your method will overcome.**
>
> Thanks for this suggestion. In the revised version, we have included a new section Problem Definition to describe the problem. In essence, we are interested in learning low dimensional latent node representations which can be trained based on the task in hand. We have added Figure 1 in the revised version illustrating this.
>
> **(ii) A banner diagram at the top of page 2 that visually describes the problem and solution. I think Figure 1 is a good starting point for this, but I would like to see it enriched with an illustration of the method (if possible). This diagram should prime the reader for the material that is subsequently presented.**
>
> We have added a new diagram (Figure 1) in the revised version that explains the pipeline of our signed GNN based methods. In the Figure, our contribution is the spectral filtering operation in each signed GCN layer. Then later in the Preliminaries section, we identify the issues with existing spectral (unsigned) methods and illustrate it in Figure 2.
>
> **(iii) I would like a table of notation added as the first section in the supplement, since the paper is very notationally heavy.**
>
> Thanks for the suggestion. We have included a notations table in Appendix A of the revised version.
>
> **(iv) Add introduction of core terminology and ideas.**
>
> We have changed the Preliminaries section significantly to address this.
>
>
> **Minor Weaknesses / Typographical Comments**
>
> We have placed all the figure and tables at the top of the corresponding pages. We have removed all the footnotes and incorporated them in the running text. We have also corrected format for equation reference. $\mathcal{V}$ is the set of nodes in graph $\mathcal{G}$.

---

### Review · Reviewer_fhAS · 2022-10-28

**Summary Of Contributions:**

The submission extends Graph Convolutional Networks, Frequency Adaptation Graph Convolutional Networks, and Magnetic Laplacian to signed graphs. The main contribution is incorporating signed laplacian in these algorithms. Experiments show that the algorithm based on the extension of magnetic laplacian generally outperforms baselines on node classification and link sign prediction tasks on signed graphs.

**Audience:**

Yes

**Claims And Evidence:**

Yes

**Requested Changes:**

See previous section

**Strengths And Weaknesses:**

Disclaimer: I have not worked on Graph Neural Networks.

The contributions in this paper are incremental and the algorithms are similar to previous work other than the use of signed laplacian. This is not a problem with TMLR. I did not find incorrect claims or glaring issues with the methodology in this paper.

These are my main questions:

1. I had a hard time understanding the experiment in 5.2. What is SNEA is this experiment and is there any intuition why the proposed method works better in high levels of imbalance?

2. Is there any evidence that the problem of negative eigenvalues actually happens to the baselines in the real-world datasets?

3. What is the connection between the possibility of negative eigenvalues and the final performance of the algorithm? Is there any previous theoretical bound that assumes nonnegative eigenvalues or one that deteriorates when some eigenvalues are negative?

---

> ### Author Response · Authors · 2022-12-28
> **Response to Reviewer fhAS**
>
> Thank you very much for the comments and suggestions. We have updated the manuscript accordingly. The major changes we have made are as follows.
> 1. We have revised the Introduction, Related Work, and Preliminaries sections significantly.
> 2. We have included Problem Definition section in the main text and notations table in Appendix.
> 3. We have included the Laplacian spectrum of all the real-world datasets in Appendix.
>
> **1. I had a hard time understanding the experiment in 5.2. What is SNEA is this experiment and is there any intuition why the proposed method works better in high levels of imbalance?**
>
> SNEA [1] is another GNN based method for representation learning over signed graphs. It is an extension of SGCN incorporating learnable attention coefficients for aggregating balanced and unbalanced paths. We considered only 2 known node labels per cluster for a total of 1000 nodes and used binary cross-entropy loss based on the known labels. We have clarified it in the revised version (Section 6.2). We have also compared our methods against SNEA for real-world node classification and link sign prediction tasks (Tables 2, 3 and 4 in the revised version).
>
> The existing methods such as SNEA and SGCN are based on balance theory and consider balanced and unbalanced sets separately in the feature aggregation process. The definitions of balanced and unbalanced sets are then extended to unbalanced graphs. In contrast, our methods are based on spectral filtering and are able to aggregate the features coming from negative as well as positive links together. Although a balanced graph has a nice spectral properties as analyzed in [2], there exist little to no work exploring the relation between the imbalance level and the spectrum of signed graphs. It is an interesting problem to explore the relationship between balance theory and
> spectral analysis for unbalanced graphs. We have mentioned this in the last paragraph of Related Work of the revised version.
>
>
>
> **2. Is there any evidence that the problem of negative eigenvalues actually happens to the baselines in the real-world datasets?**
>
> Yes, all the datasets used in the paper have negative eigenvalues (of the traditional unsigned Laplacian $\\mathbf{L}$). Moreover for all the datasets, it is not possible to normalize the Laplacian due to presence of zero degree values. In the revised version, we have plotted the histogram of the spectrum (eigenvalues) in Figure 6 of the Appendix.
>
> **3. What is the connection between the possibility of negative eigenvalues and the final performance of the algorithm? Is there any previous theoretical bound that assumes nonnegative eigenvalues or one that deteriorates when some eigenvalues are negative?**
>
> All the datasets used in the paper have negative eigenvalues (of the traditional Laplacian $\\mathbf{L}$). We could not find a relationship between the presence of negative eigenvalues (shown in Figure 6 of the revised version) and the final performance of our methods.
>
> We are not aware of any bounds relating the existence of negative eigenvalues and the performance of the signed GNNs.
>
>
> [1] Yu Li, Yuan Tian, Jiawei Zhang, and Yi Chang. Learning signed network embedding via graph attention. In Proceedings of the AAAI Conference on Artificial Intelligence, volume 34, pp. 4772–4779, 2020.
>
> [2] Thomas Dittrich and Gerald Matz. Signal processing on signed graphs: Fundamentals and potentials. IEEE Signal Processing Magazine, 37(6):86–98, 2020.

---

> > ### Comment · Reviewer_fhAS · 2023-01-17
> > **Decision**
> >
> > Thanks for the clarifications and revisions. I read the rebuttal and other reviews and responses and will vote for acceptance.

---

### Review · Reviewer_HEUa · 2022-12-18

**Summary Of Contributions:**

This paper introduces new graph neural networks for signed graphs based on their spectral analyses, where one graph neural network mostly considers low frequency as in graph convolutional network, while the other graph neural network considers both low and high frequencies with attention mechanisms. Also, the authors extend their signed graph neural networks to signed directed graphs. The authors evaluate their methods on node classification and link prediction tasks for singed graphs, showing performance improvements against baselines.

**Audience:**

Yes

**Broader Impact Concerns:**

The authors explicitly mention the broader impacts in Section 6, and I do not have any concerns about them.

**Claims And Evidence:**

Yes

**Requested Changes:**

### Suggestions and Questions
* I am wondering, the attention mechanism for considering both low- and high-frequency information could be applied to Singed-MagNet.
* In Section 5.2, it is unclear which is flipped to generate varying levels of imbalance, when experimenting with synthetic graphs.
* In Table 4, it is better to denote whether the left or the right column for each dataset is Macro-F1 or Micro-F1.
* In link prediction experiments, why the proposed methods are outperformed by existing methods on Slashdot and WikiRfA datasets?
* In Section 6, there is a typo: different signs signs.

**Strengths And Weaknesses:**

### Strengths
* The consideration of signed graphs with graph neural networks based on their spectral analyses is novel and interesting.
* This paper is extremely well-written and easy to follow.
* The quality of illustrations for explaining their signed graph neural networks is high.
* The proposed methods outperform relevant baselines.

### Weaknesses
* I don't see any major weaknesses.
* Please see minor improvements, in the requested changes section below.

---

> ### Author Response · Authors · 2022-12-28
> **Response to Reviewer HEUa**
>
> Thank you very much for the comments and suggestions. We have updated the manuscript accordingly. The major changes we have made in the manuscript are as follows.
> 1. We have revised the Introduction, Related Work, and Preliminaries sections significantly.
> 2. We have included Problem Definition section in the main text and notations table in Appendix.
> 3. We have included the Laplacian spectrum of all the real-world datasets in Appendix.
>
> **The attention mechanism for considering both low- and high-frequency information could be applied to Singed-MagNet?**
>
> Yes, the attention mechanism can also be applied to directed signed graphs as well. Since the normalized magnetic signed Laplacian $\\bar{{\\mathbf L}}^q\_{\\mathrm{n}}$ has real eigenvalues in the range of $[0,2]$, similar to Spectral-Signed-GCN-II, we can easily define low pass filter as $\\mathbf{P}^{\\mathrm{Low}} = 2 \\mathbf{I} - \\bar{{\\mathbf L}}^q_{\\mathrm{n}}$ and high pass filter $\\mathbf{P}^{\\mathrm{High}} = \\bar{{\\mathbf L}}^q\_{\\mathrm{n}}$. The only difference is going to be that the embeddings in each layer will be complex due to the presence of complex $\mathbf{A}^q$ terms in the low as well as high pass filters. Taking similar steps as in Spectral-Signed-GCN-II, the aggregation term takes the form
>
> $\\mathbf{h}^{(\\ell)}\_i =   \\mathbf{h}_i^{(\\ell - 1)} + \\sum\_{j \\in \\mathcal{N}\_i}   \\frac{\\beta\_{ij}} {\\sqrt{\\bar{d}\_i\\bar{d}\_j }} \\mathbf{A}^q(i,j) ~\\mathbf{h}\_j^{(\\ell - 1)}$
>
> Note that here $\\bar{d}\_i = \\sum\_{j=1}^{N}|\\mathbf{A}\_s(i,j)|$ as described in Section 5.1 and the attention coefficients  can be parameterized by $\\beta_{ij} = \\mathrm{tanh}(\\mathbf{a}^T[\\mathbf{h}\_i, \\mathbf{h}\_j])$ (we concatenate real and imaginary parts together). In the revised version, we have added a remark on this in Section 5.2. We implemented this  attention based method for directed signed graphs and the node classification performance with 1\% known labels is shown below.
>
> Dataset | WikiEditor (1%) | WikiElection (1%) | WikiRfA (1%)
> -------|--------|-------|-------
> Signed-MagNet | 81.0 $\pm$ 1.1   | 97.2  $\pm$ 0.6 | 96.0  $\pm$ 0.4
> Signed-MagNet (attention)   | 80.9 $\pm$ 1.0   | 96.8  $\pm$ 0.7 | 95.7  $\pm$ 0.6
>
> **In Section 5.2, it is unclear which is flipped to generate varying levels of imbalance, when experimenting with synthetic graphs.**
>
> We flip both inter-cluster as well as intra-cluster edge signs with probability $q$ for creating varying levels of imbalance. We have clarified it in the description of the corresponding section in the revised version.
>
>
> **In Table 4, it is better to denote whether the left or the right column for each dataset is Macro-F1 or Micro-F1.**
>
> In the revised version, we have mentioned it in the caption of the corresponding Table.
>
>
> **In link prediction experiments, why the proposed methods are outperformed by existing methods on Slashdot and WikiRfA datasets?**
>
> SLF is the best performing method for Slashdot and second best for Wiki-Rfa datasets. However, the SLF method is outperformed by GNN based methods in the rest of the datasets. SLF is a traditional signed graph embedding method that is not trained in an end-to-end manner. In contrast, our proposed methods have better performances against many existing GNN based methods, trained end-to-end based on the task in hand (node classification or link sign prediction).
>
> We also tried to explore the spectrum of all the datasets (see Figure 6 of the revised version) and its effect on the performance of our methods. However, we could not find a connection. The results are empirical, but are backed by interpretable spectral domain analysis of the underlying signed graphs. It is an interesting problem to explore the relationship between balance theory and spectral analysis for unbalanced (real-world) graphs. We have mentioned this in the last paragraph of Related Work of the revised version.
>
>
>
> **In Section 6, there is a typo: different signs signs.**
>
> We have corrected it.

---

> > ### Comment · Reviewer_HEUa · 2023-01-09
> > **Thanks**
> >
> > Thank you for reflecting my suggestions, and I do not have any more concerns. Best of luck.

---

### Decision · Action_Editors · 2023-01-30

**Recommendation:** Accept as is

**Comment:**

This paper studies extensions of the basic GNN architecture based on low-pass filtering (GCN) to signed graphs, that is, whose edges may be either +1 or -1. While several other architectures exist to process such graphs, the current manuscript derives an interesting alternative with good results.

**Audience:**

This paper should be of interest to the broad Graph Representation Learning community.

**Claims And Evidence:**

The paper presents clearly stated claims that are supported by clear evidence, as agreed by all three reviewers.